# Aberrant NLRP3 Inflammasome Activation Ignites the Fire of Inflammation in Neuromuscular Diseases

**DOI:** 10.3390/ijms22116068

**Published:** 2021-06-04

**Authors:** Christine Péladeau, Jagdeep K. Sandhu

**Affiliations:** 1Human Health Therapeutics Research Centre, National Research Council Canada, 1200 Montreal Road, Ottawa, ON K1A 0R6, Canada; christine.peladeauladouceur@nrc-cnrc.gc.ca; 2Department of Biochemistry, Microbiology and Immunology, University of Ottawa, 451 Smyth Road, Ottawa, ON K1H 8M5, Canada

**Keywords:** muscular dystrophy, amyotrophic lateral sclerosis, neuroinflammation, innate immune system, immune cells, astrocytes, microglia, macrophages, cytokines, therapy

## Abstract

Inflammasomes are molecular hubs that are assembled and activated by a host in response to various microbial and non-microbial stimuli and play a pivotal role in maintaining tissue homeostasis. The NLRP3 is a highly promiscuous inflammasome that is activated by a wide variety of sterile triggers, including misfolded protein aggregates, and drives chronic inflammation via caspase-1-mediated proteolytic cleavage and secretion of proinflammatory cytokines, interleukin-1β and interleukin-18. These cytokines further amplify inflammatory responses by activating various signaling cascades, leading to the recruitment of immune cells and overproduction of proinflammatory cytokines and chemokines, resulting in a vicious cycle of chronic inflammation and tissue damage. Neuromuscular diseases are a heterogeneous group of muscle disorders that involve injury or dysfunction of peripheral nerves, neuromuscular junctions and muscles. A growing body of evidence suggests that dysregulation, impairment or aberrant NLRP3 inflammasome signaling leads to the initiation and exacerbation of pathological processes associated with neuromuscular diseases. In this review, we summarize the available knowledge about the NLRP3 inflammasome in neuromuscular diseases that affect the peripheral nervous system and amyotrophic lateral sclerosis, which affects the central nervous system. In addition, we also examine whether therapeutic targeting of the NLRP3 inflammasome components is a viable approach to alleviating the detrimental phenotype of neuromuscular diseases and improving clinical outcomes.

## 1. Introduction

Neuromuscular disorders comprise of a wide range of diseases that are often classified according to the affected regions of the neuromuscular system. For instance, myopathies are diseases in which the muscle is primarily affected (e.g., muscular dystrophies and congenital myopathies), and neurogenic atrophies are diseases in which the motor neurons are affected (e.g., spinal muscular atrophy and amyotrophic lateral sclerosis). Neuromuscular disorders vary in their etiology, age of onset and disease progression; however, all neuromuscular diseases elicit progressive muscle weakness [1]. Skeletal muscle is a secretory organ that releases cytokines (also called myokines) which act as signaling messengers between the muscle and other organs in the endocrine system; thus, it plays an important role both in physiological and pathological processes [2]. Healthy muscles are subjected to consistent mechanical injury, followed by muscle degeneration and activation of acute inflammatory responses. In fact, at the early stage of muscle regeneration, following muscle damage, infiltrating immune cells (i.e., mast cells and neutrophils) clear the damaged myofibers and secrete cytokines to recruit macrophages, to regulate the inflammatory response. This cascade of events stimulates satellite cell activation, proliferation and differentiation to myotubes, resulting in mature muscle fibers [3]. The initial inflammatory response is thought to play an important role in the timely repair of the injured muscle; however, dysregulation of inflammatory signaling is harmful to the muscle and has emerged as a crucial link in the onset of muscle wasting. The activation of the transcription factor nuclear factor-kappa B (NF-κB) has been found to be a critical factor involved in the secretion of proinflammatory cytokines such as interleukin (IL)-6 and tumor necrosis factor-α (TNF-α). These cytokines are known to induce muscle wasting by stimulating proteolysis, thereby tipping the fulcrum towards protein degradation, leading in turn to impairment of muscle regeneration [4]. Interestingly, therapies that inhibit IL-6 or TNF-α and aim to reduce inflammation in patients with cachexia (weight loss and muscle wasting that occurs in chronic diseases) showed limited benefits in clinical trials [5,6,7]. This suggests that other inflammatory players may be involved in muscle wasting. Thus, recently the focus has been switched to understanding the role of the nucleotide-binding oligomerization domain (NOD)-like receptors (NLRs) with pyrin domain 3 (NLRP3) inflammasome in muscles and their connection to muscle degeneration and disease.

NLRs are a family of conserved cytoplasmic pattern recognition receptors that act as immune sensors of the innate immune system. NLRs are multiprotein oligomers capable of activating an inflammatory response; amongst them, NLRP1, NLRP3, NLRP6, NLRP7, NLRP12, NLRC4 (NLR family CARD domain-containing protein 4) and NAIP (neuronal apoptosis inhibitory protein) accomplish this via the formation of inflammasomes [8]. The stimulation and assembly of inflammasomes induces a chain-reaction of caspase 1-dependent proteolytic cleavage, along with the maturation and secretion of proinflammatory cytokines, namely, IL-1β and IL-18, resulting in a highly inflammatory form of programmed cell death, distinct from apoptosis, called pyroptosis (further described below). This process leads to plasma membrane rupture and the release of proinflammatory intracellular contents, including inflammasome components, into the extracellular milieu to promote chemotaxis and infiltration of innate immune cells at the sites of tissue damage [9].

## 2. NLRP3 Inflammasome

The NLRP3 inflammasome is one of the most studied and characterized inflammasomes due to its unique and wide diversity of stimuli, and its suggested involvement in a variety of disorders [10,11,12]. Although the NLRP3 inflammasome is important for detecting microbial and host-derived danger signals and promoting tissue repair, its over-activation may lead to severe impairments (i.e., swelling, tissue damage, internal bleeding and respiratory disablement) [12]. In this review we focus primarily on the involvement of the NLRP3 inflammasome in neuromuscular diseases.

### 2.1. Components of the NLRP3 Inflammasome

The NLRP3 inflammasome is crucial for initiating innate immune responses [13]. The stimulation, assembly, and proper functioning of the inflammasome are dependent on a variety of key players. Many excellent reviews describe in detail the fundamental steps of NLRP3 inflammasome formation and activation [9,14,15,16]. Therefore, only a brief summary is provided below.

The NLRP3 inflammasome is composed of a pyrin domain (PYD) at its N-terminus, a central NACHT domain (consisting of seven motifs which include a nucleotide adenosine triphosphate/guanosine triphosphate (ATP/GTPase) P-loop, and Walker A and B binding sites) and nine C-terminal leucine rich repeats (LRR) [17,18]. While the NLRP3 protein is maintained in an inactive monomer state endogenously, its activation promotes NLRP3 inflammasome assembly via interaction with the apoptosis-associated speck-like protein containing a caspase recruitment domain (CARD) (ASC), an adaptor protein which contains a PYD domain complementary to NLRP3 [18]. It is also well established that the NACHT domain is critical for NLRP3 function and self-association [19]. Oligomerization of NLRP3 by ASC recruits procaspase-1, which binds the CARD domain of ASC [20]. The inflammasome is thus formed of NLRP3-ASC-procaspase-1 complexes interacting together to form a ring-like structure (Figure 1). It has been suggested that the NLRP3 inflammasome may be formed of 11-subunits similar to the NLRC4 disk structure [18,21]. The assembly of inflammasome is initiated by recruitment of procaspase-1 and proximity-induced oligomerization and autoactivation, resulting in caspase-1 cleavage and release of active caspase-1. Consequently, caspase-1 cleaves pro-interleukin 1-beta (pro-IL-1β) and pro-interleukin 18 (pro-IL-18) into biologically active mature forms of IL-1β and IL-18. In addition, inflammasome-mediated programmed cell death, known as pyroptosis, is provoked following cleavage of gasdermin-D (GSDMD) by caspase-1 [22]. Activated GSDMD binds to the cell membrane and forms pores, followed by cellular swelling, plasma membrane rupture and release of cellular contents into the extracellular milieu [22,23]. This cascade of events stimulates an immune response, activates lymphocytes, and promotes a chemotaxis reaction.

Although ASC has been identified as a key adaptor that links the PYD-domain-containing NLR receptors, such as NLRP3, to the pro-caspase-1 via its PYD and CARD domains [20], it is also responsible for the formation of ASC specks (called pyroptosomes) [24]. The PYD–PYD self-association drives dimerization of ASC and permits further assembly of supramolecular insoluble aggregates independently of caspase-1 [24]. On the other hand, the CARD domain of ASC plays an important role in linking individual ASC filaments to create dense specks [25]. In addition, these specks have been shown to amplify and perpetuate inflammasome signaling by creating many caspase-1 activation sites [25].

### 2.2. Assembly and Activation of the NLRP3 Inflammasome

The NLRP3 inflammasome can be activated in response to a variety of pathogenic and environmental stimuli. In fact, members of the NLR family, including NLRP3, PYRIN and PYHIN family proteins (recognition receptors such as absent in melanoma 2 (AIM2), recently reported to play roles in cell cycle, tumor suppression and transcriptional regulation), have all been shown to assemble inflammasomes in response to cytosolic pathogen-associated molecular patterns (PAMPs) and damage-associated molecular patterns (DAMPs) [9,26,27]. These harmful stimuli include lipopolysaccharides (LPS), flagellin, viral DNA and RNA, ion fluxes, changes in pH, extracellular ATP, reactive oxygen species (ROS) and uric acid crystals [28]. The exact mechanism by which NLRP3 is activated by a vast array of pathogenic stimuli is unknown, and future research is needed to uncover the mechanisms involved.

Generally, the activation of NLRP3 inflammasome requires two types of signals: an initial priming signal, followed by an activating signal. During the priming step, production of components of the NLRP3 inflammasome is triggered by PAMPs and DAMPs. In fact, transcriptional regulation of NLRP3, pro-IL-1β and pro-IL-18 is mediated by NF-κB via, toll-like receptors (TLR) and nucleotide-binding oligomerization-domain-containing protein 2 (NOD2), or via cytokine activation (using, e.g., TNF-α) [29,30]. The other priming step involves post-translational modifications of the NLRP3 inflammasome [31]. Among these, phosphorylation, sumoylation, ubiquitination, alkylation, acetylation and nitration have been linked to NLRP3 activation in most cell types [15]. The second signal, being the activation signal, varies significantly and is stimulated by PAMPs and DAMPs such as extracellular ATP via the ligand-gated ion channel belonging to the purinergic type 2 (P2X7) receptor [32,33]. Many signals are triggered by potassium efflux (K^+^) and other ionic fluxes (i.e., Ca^2+^ and Cl^−^), uric acid crystals or particles (asbestos) causing lysosomal disruption; these are signals causing mitochondrial damage resulting in the release of ROS and mitochondrial (mt) DNA, and changes in the trans-Golgi network [14]. The two-step activation of NLRP3 is referred to as the “canonical” signaling pathway and depends on caspase-1. Although it is well established that these two steps are required to activate the NLRP3 inflammasome, it has been reported that human monocytes can promote the release of IL-1β in a NLRP3-dependent manner, in response to the priming signal alone by LPS released from gram-negative bacteria, such as *Escherichia coli* [34]. This single step activation of NLRP3 is referred to as the “non-canonical” signaling pathway and is dependent on caspase-11 in mice and caspase-4 and caspase-5 in humans [35].

### 2.3. Regulation of the NLRP3 Inflammasome

Emerging evidence indicates that protein–protein interactions are vital to NLRP3 function. Schmid-Burgk et al. have recently identified the serine-threonine kinase NIMA-related kinase 7 (NEK7), using a genome-wide CRISPR screen, as a central player in NLRP3 inflammasome regulation [36]. Interestingly, priming signaling by LPS increases NLRP3–NEK7 interactions. This distinct link between NLRP3 and NEK7 is an electrostatic complementarity interaction caused by NLRP3 being overall negatively charged while NEK-7 is positively charged [18]. NLRP3 and NEK7 can form a large oligomeric complex that leads to inflammasome assembly, as NEK7 forms interactions between adjacent NLRP3 subunits. NEK7 and NLRP3 interaction proves to be essential for: ASC speck formation, caspase-1 activation, IL-1β release and pyroptosis in vivo and in vitro [18]. Studies also show that K^+^ efflux may act as an essential trigger for NLRP3–NEK7 interaction. On the other hand, others have shown that ROS is also required [37].

The NLRP3 inflammasome has also been shown to bind other PYD containing proteins such as pyrin-only proteins (POPs), a family of proteins consisting of single PYD domains. These proteins are small cytoplasmic decoy proteins that regulate the inflammasome by either activating or inhibiting key players of the inflammasome [38,39,40]. Of the four POP isoforms (POP1, POP2, POP3 and POP4), POP1 (similar to ASC PYD) and POP2 (which is similar to NLRP3 PYD) can prevent NLRP3 inflammasome formation by either inhibiting ASC self-polymerization or inhibiting NLRP3’s interaction with ASC [38,41]. However, POPs have been shown to regulate NF-κB and may induce priming of the inflammasome [42]. Thus, POPs acts as part of a negative feedback mechanism permitting host defense and early inflammation, but also resolves inflammasome events in the long-term [41]. Card-only proteins (COPs) have also been identified in humans as regulators of the inflammasome [39]. Like POPs, COPs can both inhibit and activate the inflammasome. The three COPs isoforms, CARD16, CARD17 and CARD18, can bind to pro-caspase-1 to prevent inflammasome formation. CARD16 may also play a role in priming by activating NF-κB through receptor-interacting-serine/threonine-protein kinase 2 (RIP2), and further experiments are needed to confirm this [43]. Thus, similarly to POPs, when inflammasome assembly is initiated, COPs induce a negative feedback mechanism to regulate IL-Iβ release [39].

## 3. NLRP3 Inflammasome in Muscles

The NLRP3 inflammasome is expressed in the muscles of humans and mice. In fact, it is present in the myofibers of wild-type (WT) control mice in the tibialis anterior (TA) and is untraceable in NLRP3-knockout (KO) mice. Additionally, following a 24-h treatment with LPS, NLRP3 was increased and detected in clusters in the sarcoplasm of muscle fibers from WT mice [13]. In addition, primary skeletal muscle cells contain TLR receptors (TLR-2, TLR-4) and the P2X7 receptor, two vital players for priming and activating the inflammasome signaling pathway. Skeletal muscle cells also have the ability to secrete IL-1β in response to treatments with LPS and benzylated ATP, suggesting that muscle cells, such as immune cells, can participate in inflammasome formation [44]. NLRP3 inflammasomes have also been shown to be active in myocytes and are upregulated in degenerating muscles, muscle atrophy, muscle loss and myopathies [13,44,45]. Therefore, the potential role and mechanism of NLRP3 inflammasome in muscle wasting warrant further investigation.

A recent study from Liu et al. described the role of NLRP3 in muscle wasting and atrophy via angiotensin II (ang II) mechanistic signaling. Ang II normally plays a role in the regulation of the kidneys via retaining sodium and losing potassium, and affects blood flow by stimulating the adrenal cortex [46]. Ang II expression levels are increased in patients suffering from chronic kidney disease or heart failure who display symptoms of muscle loss and muscle wasting [47,48]. Ang II treatment of C2C12 skeletal muscle myotubes induced a dose-dependent effect on the activation of the NLRP3 inflammasome through activation of its downstream components, i.e., ASC, caspase-1, IL-1β and IL-18 [47]. In parallel, this treatment inhibited the PI3K/AKT/mTOR signaling pathway and increased the levels of major atrogene markers, atrogin-1, muscle RING-finger protein-1 (MuRF-1) and myostatin. WT mice treated with Ang II presented a muscle wasting and weight loss phenotype along with decreased muscle performance, which was reverted in the NLRP3-KO mice [47]. The Ang II-induced NLRP3 inflammasome activation was also described to be mediated through mitochondrial ROS and mitochondrial dysfunction. In fact, treatment with antioxidant MitoTEMPO, a mitochondrion-targeted superoxide dismutase mimetic that scavenges superoxide anions, inhibited the detrimental effects of Ang II in skeletal muscle [47]. Another study has demonstrated that IL-1β increases in muscle can also stimulate the expression of atrophy markers, MuRF1 and atrogin-1, supporting the involvement of inflammasomes in muscle atrophy [49,50,51]. In contrast, endogenous ASC is not expressed in zebrafish muscle; however, transgenic expression of ASC induces ASC speck assembly in these cells [52].

More importantly, mutations in the *NLRP3* gene (coding for cryopyrin) are associated with a group of clinically distinct disorders known as cryopyrin-associated periodic syndromes (CAPS) and encompass a spectrum of phenotypes described as familial cold autoinflammatory syndrome, Muckle–Wells syndrome and neonatal-onset multisystem inflammatory disease. These diseases are characterized by systemic inflammation with elevated levels of proinflammatory mediators; fever; blood neutrophilia; and increased neutrophil infiltration in the skin, joints, muscles, and cerebrospinal fluid. Many CAPS patients display musculoskeletal and neurological manifestations, including arthralgia, arthritis, myalgia and amyloidosis [53]. CAPS is caused by gain-of-function missense mutations in the *NLRP3* gene leading to aberrant NLRP3 inflammasome activation and overproduction of IL-1β [54]. As described above, activation of the NLRP3 inflammasome typically requires two signals; however, in CAPS patients, the gain-of-function decreases the activation threshold to only one signal [55]. Increased serum levels of extracellular oligomeric ASC have been found in patients with active CAPS, but not in patients with other inherited autoinflammatory diseases [56]. Hence, it is clear that the NLRP3 inflammasome is a critical innate immune sensor and that the dysregulation of this pathway has implications for the functioning of the neuromuscular system in CAPS patients.

Due to this convincing evidence connecting the NLRP3 inflammasome to muscle wasting, investigators have spent considerable efforts on determining whether NLRP3 inflammasome signaling is also involved in neuromuscular and neurodegenerative diseases associated with muscular dysfunction. The involvement of NLRP3 in these disorders is discussed below.

### 3.1. Dystrophies

#### 3.1.1. Duchenne Muscular Dystrophy and Inflammation

Duchenne muscular dystrophy (DMD) is a disorder caused by deletions or mutations in the dystrophin gene, preventing the production of functional dystrophin protein in skeletal muscle fibers, the diaphragm and the heart [57,58,59]. DMD affects males exclusively, and the birth prevalence of this fatal disease has been shown to range from approximately 16 to 19 per 100,000 live male births [60]. These patients suffer from severe muscle degeneration which results in muscle weakness, respiratory impairment, and cardiomyopathy. Loss of ambulation arises at 12 years of age and progresses until the age of 16–18 years, after which ambulation is lost [61,62]. With multidisciplinary care, including ventilatory support and cardioprotective management, many patients with DMD can live into their fourth decade of life [63]. For most DMD patients, death occurs due to respiratory or cardiac complications [64,65].

In muscles, dystrophin binds the dystrophin-associated protein complex (DAPC) at the muscle membrane and the actin cytoskeleton, which provides muscle membrane stability [66]. The absence of dystrophin causes a devastating domino effect of secondary symptoms which strongly increase the severity of DMD. These pathologies cause a chain-reaction of detrimental symptoms in DMD patients. Due to the membrane instability and defective regeneration of myofibers, extracellular calcium (Ca^2+^) influx is permitted into the cytoplasm, which can activate the NF-κB-mediated inflammatory pathway [2,67]. Activation of NF-κB is known to increase the expression of cell adhesion molecules such as intercellular adhesion molecule-1 (ICAM-1), vascular cell adhesion molecule (VCAM-1) and E-selection, leading to leukocyte adhesion and transmigration to the sites of inflammation [68]. Accordingly, atypical increases of immune cells (e.g., neutrophils, T lymphocytes) and inflammatory macrophages arise in the muscle, provoking chronic inflammation, necrosis and replacement of the muscle by connective tissue [2,69,70] (Figure 2). Consequently, research efforts focused on reducing inflammation, calcium influx, oxidative stress and fibrosis are promising approaches to improving the dystrophic phenotype [2,71]. In fact, a clinical trial is presently recruiting DMD patients to study endomysial fibrosis, deregulated inflammatory responses and Ca^2+^ influx dysfunction in order to better understand the relationship between each secondary pathology in dystrophin-deficient humans (ClinicalTrials.gov, trial NCT01823783). Other possible therapeutic strategies aimed at targeting signaling pathways to improve functional outcome of dystrophic muscles can be designed to target endogenous genes to restructure the DAPC, and tackle inflammation, calcium imbalance, fibrosis and oxidative stress [2,72,73].

Glucocorticoid therapy is the main intervention used for the treatment of DMD. Prednisone (0.75 mg/kg/day) and deflazacort (0.9 mg/kg/day) are the most commonly prescribed glucocorticoids for DMD [61]. Glucocorticoids have anti-inflammatory effects, in part, by suppressing NF-κB and activator protein-1 (AP-1), which inhibits coactivator molecules responsible for acetylating histones involved in switching on transcription of inflammatory genes [74]. Glucocorticoids also improve muscle strength in DMD patients; however, the molecular pathways controlling the beneficial and detrimental effects are not yet understood [75,76,77]. A double-blinded clinical trial comparing the benefits and risks of both deflazacort and prednisone is presently ongoing [78]. This indicates that alternate anti-inflammatory drugs should be further investigated for DMD. Such drugs explored for this disease include vamorolone (VBP15) and nonsteroidal anti-inflammatory drugs (NSAIDs). A preclinical study demonstrated the beneficial effects of the anti-inflammatory drug VBP15 on the dystrophic phenotype of mdx mice independently of undesired hormonal, growth or immunosuppressive effects often seen in glucocorticoid treatment [79]. As an alternative, NSAIDs were studied in DMD models. NSAIDs are nonselective inhibitors of the enzymes known as cyclo-oxygenase (COX) and inhibit both COX-1 and COX-2. These enzymes induce production of prostaglandins and lipid autacoids from arachidonic acid, leading to an inflammatory response [80]. Mdx mice were treated for 8–11 weeks with three types of NSAIDs—namely, aspirin and ibuprofen, which are non-selective COX inhibitors, and parecoxib, a selective COX-2 inhibitor—and scored for markers of muscle fitness and inflammation. The results from this study showed that the NSAIDs improved inflammation, fiber necrosis and muscle morphology, and aspirin in particular ameliorated resistance to muscle fatigue [81]. Another NSAID, celecoxib (Celebrex), markedly ameliorated muscle fiber integrity, improved muscle function and decreased immune cell infiltration in mdx mice [82]. These studies highlight that targeting inflammation in DMD patients is a promising approach to improving deleterious symptoms of this disease.

#### 3.1.2. NLRP3 Inflammasome Activation in Duchenne Muscular Dystrophy

As described above, inflammation contributes significantly to the pathogenesis of DMD. Although inflammation is a key component of muscle repair, chronic inflammation results in progressive muscle degeneration and weakness. Similarly to other myopathies, the NLRP3 inflammasome is upregulated in dystrophic mdx muscles and human DMD muscle cells [13,83] (Figure 3). In parallel, expression levels of the components of the NLRP3 inflammasome cascade, such as adaptor protein ASC, pro-caspase-1, pro-IL-1β and mature IL-1β protein, are also elevated in mdx mice myocytes compared to WT mice [83]. Upstream receptors of inflammasomes, TLRs, including TLR-1, 2, 3, 4, 7, 8 and 9, are also expressed in a variety of skeletal muscles at different levels in mdx mice [84]. Recent work has shown that ablation of TLR or the TLR adaptor protein called myeloid differentiation primary response gene 88 (myd88) in mdx mice offers benefits to their muscle [84,85]. In addition, NLRP3-KO in mdx mice induced significant improvements in grip strength, fatigue resistance and endurance compared to a control mdx mouse [13]. In line with this, knockdown of NLRP3 using lentiviral shRNA technology elicited similar performance benefits and morphological changes in mdx mice [83]. Histopathological studies have shown that abnormal fiber size distribution is a hallmark of dystrophic muscles [86,87]. Accordingly, mdx mice display abnormal proportions of small and large muscle fibers [87]. NLRP3 knockdown in mdx mice restores a more equal distribution in muscle fiber size and significantly reduces centrally nucleated myofibers (indicative of fewer regenerating fibers). Based on these findings, NLRP3 has emerged as an intriguing target for DMD research and is under investigation to alleviate skeletal muscle inflammation. Interestingly, a recent study has demonstrated that the glucocorticoid prednisone, the main treatment for DMD patients [61], inhibited NLRP3 expression, reduced inflammasome signaling and reduced related proinflammatory cytokines [88]. These findings suggest that glucocorticoid treatment in DMD patients may, in part, induce its beneficial effects through the reduction of inflammation via the NLRP3 inflammasome.

A new and promising approach to target NLRP3 involves the hormone adiponectin. In fact, this hormone is abundantly secreted by adipocytes in normal physiological conditions [89] and has recently emerged as a master regulator of inflammation in several tissues, including the skeletal muscle [13,90,91]. Furthermore, it has been demonstrated that the protective effects, in part, were mediated through the upregulation of the microRNA miR-711, and possibly by induction of anti-inflammatory pathways [92]. In contrast to control mice, transgenic mdx mice overexpressing adiponectin showed decreased inflammation and oxidative stress markers involved in dystrophic muscle damage [93]. A lack of adiponectin expression worsens the mdx pathology [94]. Along these lines, a recent publication from Boursereau et al. explored the effects of NLRP3 and adiponectin in muscle, in a DMD context [13]. Firstly, they demonstrated that NLRP3 is upregulated in dystrophic mdx TA muscle and in human DMD primary cells, which was abolished with a 24-h adiponectin treatment. This study also demonstrated that similarly to what was seen in the transgenic mdx/NLRP3-KO mice, the adiponectin-mediated decrease in NLRP3 promoted beneficial effects on the dystrophic phenotype of mdx mice. A decrease in the serum levels of the enzyme creatine kinase (CK), a marker of muscle damage, and an improvement of physical performance, were observed [13,93,94]. These findings reveal the potential involvement of NLRP3 in DMD pathogenesis and adiponectin as a potential therapeutic approach to target inflammation in DMD.

Another potential NLRP3 target characterized in dystrophic mice is ghrelin. This hormone, typically involved in regulating appetite, has been shown to induce anti-inflammatory activity in many inflammatory diseases [95,96,97]. Accordingly, a 4-week treatment with ghrelin in 4-week-old mdx mice reduced the levels of NLRP3, ASC, pro-caspase-1, cleaved caspase-1, pro-IL-1β and IL-1β, indicative of an overall decrease of the inflammasome signaling molecules. This treatment also improved motor performance as determined by in vivo behavioral testing and morphological assessment of pathology in the mdx muscles. Furthermore, the inhibition of NLRP3 inflammasome by ghrelin was partly mediated by the suppression of janus kinase 2/signal transducer and activator of transcription 3 (JAK2-STAT3) and the p38 mitogen-activated protein kinase signaling pathway [83], typically involved in modulation of inflammation, hematopoiesis, immunity, cell growth, differentiation and proliferation [98].

Although the studies discussed above highlight the importance of NLRP3 in DMD and other dystrophies (described below), contradicting results from a recent study demonstrate that genetic disruption of the inflammasome central adaptor ASC has minimal effects on the phenotype of mdx mice [99]. Therefore, further experimentation is warranted to resolve these discrepancies and delineate the mechanisms involved.

#### 3.1.3. NLRP3 Inflammasome Activation in Limb Girdle Muscular Dystrophy

Limb–girdle muscular dystrophy (LGMD) is a genetically and clinically heterogeneous family of rare progressive muscle disorders [100,101]. LGMD disorders are caused by genetic variants resulting in aberrant synthesis of proteins involved in different parts of the muscle fiber, such as the sarcolemma (muscle fiber membrane), sarcoplasm, sarcomere, nucleus and the extracellular matrix [102,103]. LGMD onset initiates at different ages; however, they are non-congenital. Patients suffering from LGMD primarily show weakness and wasting of the proximal limb’s musculature (limb-girdle area), and some cases show cardiac impairments [104]. Mutations in *dysferlin* (*DYSF*), which encodes for the dysferlin protein, is known to cause limb-girdle muscular dystrophy type 2B (LGMD2B). Dysferlin is localized at the sarcolemma of muscle fibers, but unlike dystrophin, it is not part of the DAPC [105]. Dysferlin is responsible for fusion of repair vesicles with the plasma membrane to repair the damaged muscle membrane, and has been suggested to be involved in the control of muscle inflammation [105,106,107]. Dysferlin expression has also been directly correlated with the ability of mitochondria to improve the repair of the muscle fiber membrane [108].

Mild myofiber damage in dysferlin-deficient muscle is sufficient to activate inflammatory signaling, which may contribute to the progression of the disorder. Membrane instability has been shown to increase ATP release from myofibers [109,110], and extracellular ATP activates caspase-1 and promote proinflammatory cytokine secretion [32,33]. Pro-IL-1β and pro-caspase-1 are contained in secretory lysosomes where they are released in the presence of an exocytosis-inducing stimulus, such as extracellular ATP [111]. The expression of the purinergic receptor P2X7 that binds ATP is also elevated in dysferlin-deficient mice compared to control mice. Since extracellular ATP binding to P2X7 receptor is a potent activator of NLRP3 inflammasomes, researchers further explored the role of NLRP3 in dysferlin-deficient models. Indeed, biopsies from LGMD2B patient muscle contained higher levels of protein components of the inflammasome, including NLRP3, ASC and pro-caspase 1, compared to healthy subjects. Similar outcomes were seen in the dysferlin-deficient SJL/J mice, a mouse model that develops a spontaneous myopathy phenotype resulting from a splice-site mutation in the *DYSF* gene [44,112]. Treatment of primary skeletal muscle cells from dysferlin-deficient mice with inflammasome activators such as LPS and benzylated ATP led to more secretion of IL-1β into the culture supernatant compared to untreated muscle cells [44]. Thus, with membrane fragility and decreased muscle fiber integrity, as in LGMD2B, the damaged muscle cells are more susceptible to NLRP3 inflammasome activation and the inflammatory response. Taken together, these findings suggest that there are potential therapy avenues for targeting the NLRP3 inflammasome in LGMD2B patients.

The entry of Ca^2+^ ions due to sarcolemmal tears or the activation of leaky calcium channels by sarcolemmal stretching creates calcium excesses in the muscle fibers and in the mitochondria, causing swelling [113,114]. Interestingly, it has been shown that an increase in cytosolic Ca^2+^ promotes NLRP3 inflammasome activation. In fact, one study suggested that Ca^2+^ directly regulates NLRP3 inflammasome activation by inducing interaction between NLRP3 and ASC in macrophages [115]. It is also suggested that the increase of cytosolic Ca^2+^ may provoke Ca^2+^ overloading of mitochondria, resulting in mitochondrial dysfunction that leads to NLRP3 inflammasome activation [116]. Thus, it does not come as a surprise that dystrophic muscles of LGMD2B and DMD patients that have membrane instability and increased Ca^2+^ influx could have excessive NLRP3 inflammasome activation. However, this has not been yet fully explored.

It is unclear whether inflammasome activation seen in dysferlinopathies is only due to sarcolemmal wounding of the muscle fibers or whether it is also caused by lysosomal dysfunction. Dysferlin-containing vesicles move along microtubules via kinesin motor proteins, and have been shown to fuse with lysosomes in response to membrane damage, to create large wound sealing vesicles [117]. Dysferlinopathy patient cells harbor a greater number of lysosomes and vesicles than healthy subjects [118]. In addition, lysosomal destabilization and rupture increases the production of mitochondrial ROS and NLRP3 activation, and promotes the release of IL-1β and IL-18 in both HUVEC and THP-1 monocytes [119,120]. Hence, dysferlin deficiency may result in increased stimulation of NLRP3 inflammasome via lysosomal dysfunction. The precise mechanisms involved in the activation of the NLRP3 inflammasome in dysferlinopathies warrant further investigation.

## 4. Amyotrophic Lateral Sclerosis

Amyotrophic lateral sclerosis (ALS, also known as motor neuron disease and Lou Gehrig’s disease) is an adult-onset, progressive neurodegenerative disease affecting approximately 2 per 100,000 individuals world-wide, with a higher incidence among men than women. It is characterized by the selective degeneration of motor neurons in the brain, spinal cord and skeletal muscles, leading to severe muscle atrophy and eventually paralysis [121]. The disorder initiates with focal weaknesses but spreads and affects most muscles, leading to paralysis of the whole body. Due to the large number of causative factors that are suggested to increase in risk of developing ALS, disease pathology and clinical presentation of ALS are markedly heterogeneous. The rate of disease progression is very aggressive, and death typically occurs within 2–5 years post-diagnosis, generally due to respiratory failure [122]. The vast majority of ALS cases (>90%) are sporadic, but ~5–10% of the cases are familial [123]. The pathological hallmarks of ALS include the presence of intraneuronal inclusions composed of insoluble aggregated proteins [124].

The molecular mechanisms involved in the initiation and progression of ALS remain unclear; however, mutations in the genes that are important for neuronal survival and function have been identified in patients with the familial and sporadic forms of the disease. A hexanucleotide repeat expansion (GGGGCC) mutation in the non-coding region of the non-coding chromosome 9 open reading frame 72 (*C9ORF72*) gene is the most common genetic cause of ALS patients and is present in nearly 50% of patients with familial ALS, and 5–10% of patients with sporadic ALS [125]. Although the molecular mechanisms by which the C9ORF72 protein causes neurodegeneration has not been fully elucidated, both loss-of-function through haploinsufficiency and gain-of-function through accumulation of toxic peptides have been implicated. Knockdown of C9ORF72 protein using silencing RNA dysregulated autophagy and inhibited endocytosis, supporting its role in endosomal trafficking and protein degradation [126]. Mutations in the Cu/Zn superoxide dismutase (*SOD1*) gene are another set of causes of the familial form of the disease; both gain-of-function and loss-of-function mutations have been reported [127,128]. Since the discovery of the first missense mutation in 1993 in the *SOD1* gene, advances in human genetics have identified more than 185 *SOD1* variants [129]. SOD1 is a ubiquitously expressed enzyme that is found in the cytoplasm and intermembrane space of mitochondria. It is known to bind Cu^2+^ and Zn^2+^ ions and convert highly reactive and toxic superoxide radicals to oxygen and hydrogen peroxide, considered to be the less toxic species. Hydrogen peroxide is subsequently converted to water and oxygen in the presence of catalase, thereby promoting an antioxidant defense mechanism in the cell [126]. It has been suggested that mutations in *SOD1* result in conformational and functional changes, leading to the inability of the enzyme to scavenge superoxide radicals, leading to oxidative stress. Recent progress in ALS research has shown that an *SOD1* mutation contributes to ALS pathology by causing toxic gain-of-function of the mutated protein, in turn leading to misfolding and oligomerization.

Familial ALS has also been linked to mutations in more than 50 separate genes that are known as disease-modifying or causative of ALS, of which TAR DNA-binding protein 43 kDa (TDP-43, encoded by *TARDBP*) fused in sarcoma (*FUS*) and *C9ORF72*, as described above, have been most extensively characterized. A growing body of evidence suggests that aberrant post-translational modifications, including ubiquitination, acetylation and phosphorylation, can lead to aggregation and mislocalization of these proteins, thereby resulting in cellular dysfunction [130]. For instance, TDP-43 is an RNA-binding protein that localizes in the nucleus and regulates a multitude of RNA processing steps, such as transcriptional repression, microRNA biogenesis, pre-mRNA splicing, mRNA stability, mRNA transport and translation. It has been suggested that aggregated TDP-43 shuttles from the nucleus to the cytoplasm, and its accumulation and redistribution to the neurites lead to the loss-of-function of normal TDP-43, thereby affecting cellular metabolism. Alternatively, the presence of cleaved TDP-43 C-terminal fragment(s) (25 and 35 kDa) may result in a toxic gain of function [131]. More recently, it has been shown that neuroinflammation in ALS is driven, in part, by cytoplasmic DNA sensor cyclic guanosine monophosphate (GMP)-AMP synthase (cGAS) resulting from TDP-43-induced mtDNA release. In fact, inhibition of cGAS averts upregulation of NF-κB and type I IFN responses in iPSC-derived motor neurons and in TDP-43 mutant mice [132]. Similarly, FUS is also an RNA-binding protein. It is localized in the nucleus where it regulates RNA processing pathways. In neurons, FUS is present in RNA-transporting granules, axons, dendrites, and excitatory synapses. While the loss of FUS does not trigger motor neuron degeneration and is not sufficient to cause ALS, overexpression of mutant or wild-type FUS triggers motor neuronal cell death, supporting toxic gain of function [126].

In both sporadic and familial ALS, motor neuron cell death occurs by complex mechanisms that involve dysfunction in several cellular processes, including endoplasmic reticulum and mitochondrial stress, autophagy, proteasome, oxidative stress (ROS and nitric oxide), glutamate excitotoxicity, calcium overload, aberrant RNA/DNA regulation, axonal transport system impairments and the “prion-like” propagation of misfolded protein, where the aggregated/abnormal proteins act as seeds of neuropathology [133]. Neuroinflammation is increasingly being recognized as a prominent pathological feature of ALS. Here we briefly review the role of the immune system, especially focusing on the role of the central nervous system (CNS) innate immunity in neurodegeneration, and further discuss potential therapeutic approaches to precisely dampening chronic neuroinflammatory responses in ALS by targeting the NLRP3 inflammasome.

### 4.1. Neuroinflammation and Motor Neuron Cell Death

Although the etiology and pathogenesis of ALS continues to be debated, accumulating evidence suggests that neuroinflammation is an active contributor to the initiation and progression of ALS [134]. The neuroinflammatory responses are mediated, at least in part, by the activation of resident innate immune cells, i.e., microglia, astrocytes and mast cells, in response to sterile triggers (e.g., protein aggregates). These responses are initially beneficial and protect the host. However, failure to regulate these processes leads to a state of sustained/chronic activation of the immune system, which constitutively secretes pathogenic levels of proinflammatory mediators and can contribute to motor neuron death and disease progression [135,136,137]. Indeed, reactive astrocytes and microglia are generally found to accumulate at sites of motor neuron degeneration and have been shown to release proinflammatory cytokines, including TNF-α, CD11b, IL-17A, interferon gamma (IFNγ), CD40L and IL-6; and NLRP3 inflammasome-induced cytokines IL-1β and IL-18, in post-mortem human tissues and animal models of ALS, respectively [138]. In addition, ALS patients display a detrimental neuroinflammatory phenotype, not only due to increased production of proinflammatory cytokines by activated resident immune cells but also because of increased peripheral immune cell migration into the brain, which can further aggravate inflammatory responses [139]. Increasing evidence points to an extensive and complex interaction occurring between the innate immune cells, i.e., mast cells, astrocytes and microglia, which orchestrate immune responses during neuroinflammation, and neurodegeneration. Several studies have also demonstrated increased infiltration of immune cells from the circulation into the brain; however, it is still unknown whether they are protective or detrimental [140]. Notably, blood–CNS barrier impairments and changes in endothelial cells have been documented in both human pathological samples and animal models of ALS [141].

Current evidence also points to the roles of the peripheral immune system in the initiation and progression of ALS, which have recently been reviewed [142]. A histopathological, post-mortem assessment of human spinal cord and brain tissues revealed T-lymphocyte infiltration at the sites of motor neuron cell death, suggesting engagement of the adaptive immune system [143]. Flow cytometry analysis demonstrated alterations in T-lymphocyte populations in the blood of ALS patients as compared to healthy subjects. Decreases in the regulatory T-lymphocyte (Tregs) cell counts and mRNA levels of FoxP3 (transcription factor forkhead box P3) were associated with accelerated disease progression in ALS patients [144]. In addition, Tregs obtained from ALS patients were less effective at suppressing responder T-lymphocytes as compared to Tregs from healthy subjects, and circulating monocytes of ALS patients expressed a distinct proinflammatory signature [145]. In a human clinical study involving 33 patients with sporadic ALS and 38 healthy control subjects, Sheean et al. recently showed that a systemic reduction in peripheral Tregs (including effector CD45RO^+^/FoxP3^+^ population) correlated with an enhanced rate of ALS progression. They also tested the therapeutic potential of enhancing effector Tregs in transgenic mice harboring a G93A mutation in *SOD1* (SOD1^G43A^) [146]. Expansion of effector Tregs population was associated with slower disease progression and improvement in the lifespan for ALS mice. At the histological level, marked reductions in astrogliosis and microgliosis were accompanied by increased motor neuron survival [146]. Tregs are known to suppress inflammation by inhibiting proinflammatory effector T-lymphocytes, and proinflammatory innate immune myeloid cells, such as monocytes and macrophages; and by promoting neuroprotective microglia [144]. Taken together, these studies demonstrate a clear link between patient Tregs populations, monocyte profiles and ALS progression, and support that there is active crosstalk between the innate and adaptive immune systems. Strategies aimed at enhancing Tregs might be immunomodulatory and provide neuroprotection leading to improved clinical outcomes in ALS patients.

A vast number of studies highlight the complexity of the immune responses in ALS patients and provide compelling evidence for immune dysregulation in the pathogenesis of ALS (comprehensively reviewed in [147,148]). There exists a conundrum between hyperinflammation and inefficient immune responses to pathogenic stimuli in ALS patients, possibly due to patient-specific mutations in ALS-associated genes or polymorphisms in cytokine/chemokine genes that might further affect the patient’s ability to mount appropriate immune responses.

### 4.2. Astrocytes: Stars in the Darkness

Astrocytes represent the most abundant cell type in the CNS that regulate and maintain homeostasis [149]. Such astrocytes are described as “homeostatic astrocytes” which are involved in critical CNS functions, including but not limited to the maintenance and regulation of the extracellular microenvironment, blood–brain barrier (BBB) integrity, cerebral blood flow, antioxidant and trophic factor support, uptake and recycling of neurotransmitters and detoxification of extracellular glutamate and ROS [150,151,152]. Astrocytes respond to all forms of CNS damage, including misfolded aggregated proteins composed of SOD1, FUS and TDP-43; dying neurons; and elevated ROS and cytokines, by a process known as “reactive astrocytosis” which is accompanied by increased expression of intermediate filaments such as vimentin and glial fibrillary acidic protein (GFAP) [153]. Reactive astrocytes also undergo a dramatic morphological transformation and exhibit ramifications of GFAP-positive hypertrophic processes which are associated with a broad spectrum of molecular and functional changes [154]. Liddelow et al. separated reactive astrocytes into A1 and A2 subtypes: A1 astrocytes were shown to be neurotoxic and exacerbate disease pathology, whereas A2 astrocytes secreted neurotrophic factors and promoted neuronal survival [155]. It has now been established that reactive astrocytosis is a highly dynamic process whereby some astrocytes can lose their normal functions and gain neurotoxic functions, which in turn, can negatively impact the surrounding neural tissue whereas others can revert back to their normal homeostatic states and promote neuronal survival. However, identification of the exact molecular subtypes of reactive astrocytes that are beneficial or detrimental to neuronal health has been technically challenging, and their roles in the neurodegenerative process remain elusive.

#### 4.2.1. Astrocyte Dysfunction in ALS

Although mutant SOD1 causes intrinsic damage to motor neurons, it may not be enough for the initiation and propagation of ALS pathogenesis. Reactive astrocytes have emerged as important mediators of mutant SOD1 toxicity and motor neuron death. In ALS patients, reactive astrocytes become dysfunctional and lose their beneficial functions and interactions with motor neurons [156], thereby evoking detrimental effects on the muscles. In two hallmark studies, it has been demonstrated for the first time that astrocytes carrying mutant SOD1 and cocultured with motor neurons derived either from mouse embryonic spinal cord or embryonic stem cells caused neurotoxicity [157,158]. This gain of function by astrocytes resulted in toxicity that was highly specific to motor neurons, and astrocytes expressing mutant SOD1 did not cause the death of other neuronal subtypes. Furthermore, other CNS cell types, for example, microglia, cortical neurons, and fibroblasts, carrying mutant SOD1, did not cause motor neuron cell death. Knockdown of SOD1 in astrocytes reversed the neurotoxic effect demonstrating that astrocytes carrying a SOD1 mutation can induce toxicity in neighboring motor neurons [159]. These studies revealed that astrocytes and astrocyte-specific neurotoxic factor(s) play active roles in motor neuron death in ALS.

Numerous studies have shown that the loss of function of astrocytes can also result in motor neuron death and ALS disease progression. Both sporadic and familial ALS cases have ~90% lower levels of astrocytic glutamate transporter-1 (GLT-1), which result in decreases in removal of extracellular glutamate by astrocytes and motor neuron death [160]. Similarly, we have also shown that exposure of mixed neuron-astrocyte cocultures to toxic levels of glutamate resulted in increased H_2_O_2_ production and decreased mitochondrial function, which was accompanied with neuronal loss [161]. The uptake of the excitatory neurotransmitter glutamate by astrocytes is a particularly critical astrocytic function, which regulates glutamatergic neurotransmission and is necessary to avoid excitotoxic injury and neuronal death. Synaptic homeostasis of glutamate involves its removal from the synaptic cleft via high-affinity glutamate transporters GLT-1, GLAST and EAAC1, and the glutamate-catabolizing enzyme glutamine synthase [162]. ALS spinal cord astrocytes have high levels of monoamine oxidase B (MAO-B), an enzyme located on the outer mitochondrial membrane, which is known to metabolize dopamine [163]. This increased expression of MAO-B in ALS supports the hypothesis that motor neuron death in ALS is triggered by astrocytic reaction. However, treatment with selegiline (Eldepryl), a MAO-B inhibitor, had no significant effect on the rate of clinical progression or outcome of ALS [164]. These studies point to a complex intricate interplay between motor neurons and astrocytes and suggest that astrocytes play active roles in neurodegeneration and ALS disease progression.

Activated astrocytes also secrete an array of neurotoxic factors, including ROS, nitric oxide, superoxide anions and proinflammatory cytokines (IL-1β, IL-6 and TNF-α), which might contribute to neuroinflammation and motor neuron death [165]. Nitric oxide can diffuse through cellular membranes and react with superoxide anions to form peroxynitrite, a potent oxidizing and nitrating agent. Peroxynitrite is known to cause post-translational modification of tyrosine residues in proteins to form nitrotyrosine, an indicator of peroxynitrite-mediated protein damage that can result in protein dysfunction. To this end, nitrated proteins have been detected in both human ALS patients and transgenic animal models of ALS [166]. However, caution should be exercised in the interpretation of post-mortem brain tissue data, as they only provide a snapshot of the pathology at the terminal stage of the disease.

#### 4.2.2. Astrocyte Dysfunction and the NLRP3 Inflammasome

Evermore studies are supporting the involvement of inflammasomes, including the NLRP3 inflammasome, in ALS pathogenesis. Indeed, the inflammasome has been shown to be activated by neurodegeneration-associated molecular patterns (NAMPs), such as misfolded protein aggregates, which are constitutively expressed in post-mortem human ALS tissue and the neural tissue of mice expressing mutant SOD1^G93A^ [167,168,169]. Upstream signaling molecules NF-κB and TRL-4 are also increased in SOD1^G93A^ mice [169,170]. NLRP3 inflammasome components’ (i.e., NLRP3, ASC, caspase-1 and IL-18) expression levels have been found to be increased in the spinal cord tissue, where spinal cord astrocytes were identified to be the focal NLRP3 inflammasome expressing cell type [171]. Moreover, caspase-1 is also increased in the sera of ALS patients [172].

NAMPs can also lead to inhibition of autophagy in astrocytes, in turn accelerating neurodegeneration. Autophagy is the primary cellular lysosomal degradative pathway that is involved in the degradation of damaged proteins and dysfunctional organelles. This pathway has now received recognition as limiting detrimental and uncontrolled activation of inflammasomes [173]. An accumulating body of research has demonstrated that impairments in the autophagy machinery induce abnormal activation of inflammasomes that can contribute to the development of ALS [174]. Other important factors that may mediate ALS progression are mitochondrial dysfunction and autophagy regulation. Mitochondrial dysfunction results in excessive ROS release, which is seen in sporadic ALS patients [175]. Additionally, transgenic SOD1 mice show symptoms of mitochondrial swelling and dysfunction in early stages of ALS [176]. Although the direct involvement of mitochondrial defects and NLRP3 activation has yet to be characterized in ALS, it is well established that mitochondrial DNA and phospholipid cardiolipin from damaged mitochondria activate NLRP3 [177,178]. Thus, there is a possibility that mitochondrial protectants shown to improve ALS pathology (such as GNX-4728) may promote their beneficial effects, in part, via inhibition of NLRP3 inflammasome. Interestingly, an autophagic inducer, trehalose, was also shown to have protective effects towards mitochondria in SOD1^G93A^. In addition, it improved the loss of motor neurons, SOD1 aggregation, autophagic flux and lifespan [179]. Thus, these findings support the notion that regulating autophagy may be a potential approach to treating ALS. Meissner et al. has shown that the aggregation and proinflammatory response by mutant SOD1 in the SOD1^G93A^ transgenic mice are improved by autophagy. Indeed, pharmacological inhibition of autophagy increased SOD1 in the cytosol and induced IL-1β release following SOD1^G93A^ stimulation in vitro [180]. In summary, these studies suggest that autophagy plays a key role in counteracting the cytoplasmic accumulation of mutant SOD1 and may be an important player in inflammatory signaling of inflammasomes. Although autophagy inhibition reverses disease pathology and extends the lifespan in mouse ALS models, it is noteworthy that chronic autophagy inhibition has been linked to increased cancer risk, suggesting a biphasic role of autophagy in cancer [181].

### 4.3. Microglia: The Defense System of the CNS

Microglia represent the resident innate immune cells of the CNS that are important for the maintenance of homeostasis and neuronal survival. Like other tissue macrophages, microglia are the professional phagocytes of the CNS that defend against microbial and non-microbial threats and clear debris from damaged/dead cells. As a first line of defense, following detection of NAMPs, microglia activate a defense program that arms them with a toolkit to transition from the “homeostatic” state into an “activated” or a “disease-associated” state [182]. During the transition to the activated state, microglia undergo numerous morphofunctional changes, release inflammatory mediators (TNF-α, IL-6, nitric oxide, ROS and superoxide anions) and trigger activation of the NLRP3 inflammasome [183]. Following damage resolution, microglia shift back to the homeostatic state and secrete neuronal survival molecules, such as anti-inflammatory cytokines (IL-10 and IL-4), and neurotrophic growth factors, to ensure neuronal fitness. However, chronic stimulation switches microglia to a hyperinflammatory state, resulting in continuous production of proinflammatory mediators, and leading to motor neuron death. Using high-throughput RNA sequencing, a recent study has shown that microgliosis in ALS is driven by a subpopulation of microglia with transcriptional signatures similar to those of disease-associated microglia [184]. This unique population of microglial cells might play an important role in driving neuroinflammatory responses and synaptic loss seen in ALS disease progression.

#### Microglial Dysfunction and the NLRP3 Inflammasome

A characteristic hallmark of sporadic ALS is the presence of intracellular inclusions composed of SOD1, FUS, TDP-43 and C9ORF72 protein aggregates in motor neurons and astrocytes, which can also be found in brain parenchyma. Microglia respond to these pathogenic stimuli by releasing proinflammatory mediators and activating the NLRP3 inflammasome, which is emerging as a key player in driving neuroinflammation and neurodegeneration in ALS. In this context, exposure of cultured murine and human macrophages to monomeric TDP-43—but not its fibrillary form—was shown to activate the NLRP3 inflammasome and release IL-1β. In addition, C9ORF72-derived glycine–arginine (GR) dipeptide repeats were also phagocytosed by macrophages, stimulated NLRP3 and released IL-1β [185]. This study suggests that ALS-associated proteins can drive inflammatory processes.

Accumulating evidence suggests that microglial activation precedes astrocyte reaction, and microglial factor(s) trigger astrocytes to acquire a reactive and neurotoxic phenotype. Activated microglia are observed at the early phases of disease, especially prior to the signs of disease onset, and transgenic ALS mice showed increased expression levels of inducible nitric oxide synthase, NADPH oxidase, IL-10, IL-6, TNF-α and CCL2. Although there is clear evidence of an early immune response, it is still not clear whether these responses are protective or injurious. The role of microglia in ALS disease pathogenesis has been recently reviewed [186]. Similarly to astrocytes, microglia have also been shown to contribute to glutamate excitotoxicity in the mutant SOD1 mouse model [187]. In addition, activated microglia play a predominant role in neuroinflammation in ALS through the release of TNF-α, IL-1β and oxidative stressors via NF-κB, and through inflammasome activation, as seen in SOD1^G93A^ transgenic mice [135,180,188]. Crossing the SOD1^G93A^ transgenic mice with caspase-1 or IL-1β-deficient mice prolonged the lifespan of the SOD1^G93A^ mice and ameliorated the inflammatory pathology [180]. Treatment of SOD1^G93A^ mice with a recombinant human IL-1 receptor antagonist (Anakinra) showed similar beneficial effects [180]; however, no benefit was seen for ALS patients in a clinical trial [189]. In addition, results from SOD1^G93A^ mice suggested that caspase-1 and IL-1β in microglia are produced in an NLRP3-independent manner [180]. Conversely, using NLRP3-GFP knock-in mice, a recent study by Deora et al. demonstrated microglial expression of NLRP3 [135]. Furthermore, IL-1β secretion was inhibited using MCC950, a selective NLRP3 inhibitor, confirming that indeed microglial IL-1β production arose via the NLRP3-dependent inflammasome pathway stimulated by ROS and ATP. Treatment of microglial cells obtained from SOD1^G93A^ mice with the antioxidant cyclo (His-Pro) inhibited NLRP3 inflammasome activation by reducing protein nitration and ROS levels via NF-κB inhibition [190]. Importantly, NF-κB-regulated NLRP3 inflammasome formation has also been shown to be activated in the microglia of TDP-43^Q331K^ transgenic mouse model of ALS [191]. Altogether, compelling evidence suggests that motor neuron degeneration in ALS arises from the toxic and proinflammatory effects of surrounding glial cells. Hence, there is an urgent need for the development of novel therapeutics targeting key pathways in microglial cells to inhibit NLRP3 inflammasome activity, neuroinflammation and ALS disease progression.

### 4.4. Mast Cells: Guardians of the CNS

Mast cells are present in small numbers in the healthy human brain and dura of the spinal cord, but not in the cord parenchyma. They contain numerous granules that store pre-formed or newly synthesized arsenal of proteases, histamine, proteoglycans, growth factors, immunomodulators and cytokines such as TNF-α. As opposed to microglia, mast cells are recognized as the first responders to pathogenic stimuli and respond rapidly by releasing these mediators into the microenvironments of their respective tissues. These mediators can elicit profound effects on neighboring neural cells and also recruit peripheral immune cells into the CNS, further exacerbating inflammatory responses. We highlighted the intimate and intricate molecular communication between mast cells and microglia in the context of neurodegeneration in a recent review [192].

#### Mast Cells and the NLRP3 Inflammasome

As discussed, both astrocytes and microglia have received much recognition for their role of promoting neuroinflammatory responses, especially before the onset of symptoms, and for correlating with disease progression in rodent models of ALS. However, mast cells, granulocytes derived from myeloid stem cells, have been underappreciated and understudied in ALS, and have only recently been the focus of research. In fact, autopsied quadriceps muscles from ALS patients showed a large number of infiltrating degranulated mast cells, which was not seen in healthy subjects. Mast cells and neutrophils were also found infiltrating the motor axons of SOD1^G93A^ rats [137]. In addition, mast cells have previously been identified as the main cell population responsible for IL-1β production via NLRP3 activation [193]. These findings indicate that many immune cells may be involved in the NLRP3 inflammasome-induced inflammatory pathology of ALS, and targeting NLRP3 may provide therapeutic benefits against this devastating disease.

Encouraged by the promising results obtained using cromolyn sodium in cellular and preclinical animal models of AD [194], Granucci et al. recently tested the effects of cromolyn in the SOD1^G93A^ mouse model of ALS [195]. Cromolyn sodium is a mast cell stabilizer approved by the FDA for the treatment of asthma. Interestingly, cromolyn treatment led to a decrease in the denervation of the neuromuscular junction of the tibialis anterior muscle and an increase in motor neuron survival in the lumbar spinal cord. The protective effects were mediated by reductions in proinflammatory cytokine and chemokine levels in the spinal cord and plasma, and mast cell counts; and degranulation in the tibialis anterior muscle. This study provides compelling evidence for the involvement of mast cells in the early phase of ALS disease progression.

### 4.5. CNS Innate Immune Cell Senescence

Aging is recognized as the most prevalent risk factor for neurodegenerative diseases, including ALS [196]. Age-related decline in immune system function, referred to as “immunosenescence”, has been observed in both astrocytes and microglia, rendering them more vulnerable to neurodegeneration. Typically, senescent astrocytes exhibit irreversible cell-cycle arrest; high levels of expression of cell-cycle inhibitors p16 and p21; high levels of expression of tumor suppressors such as p53, and/or pRb rendering cells resistant to apoptosis; increased expression of GFAP and vimentin; downregulation of neurotrophic growth factors; increased lysosomal activity; and most importantly, increased secretion of proinflammatory senescence-associated secretory phenotype (SASP) factors [197]. Aged astrocytes upregulate inflammatory gene expression networks, such as proinflammatory cytokines, chemokines, growth factors and proteases, characteristic of A1 reactive phenotype, suggesting perturbations in immune regulation pathways [198]. In addition, aged astrocytes also display senescence-associated markers of oxidative stress and upregulated antioxidant responses (SOD, catalase and the glutathione antioxidant system) and possess a deficiency in glutamate homeostasis. Furthermore, aged astrocytes were less able to support motor neurons which were augmented in ALS astrocytes, thereby promoting motor neuron loss [199]. In line with this, neurons cultured in the presence of senescent astrocytes showed increased vulnerability to glutamate toxicity, possibly due to decreased efficiency of the glutamate transport machinery [200]. The impacts of ALS-associated mutations in *SOD1, FUS, TARDBP* and *C9ORF72* on astrocyte senescence are not yet known. Similarly to astrocytes, more activation of inflammatory pathways has also been documented in microglia during normal aging [198]. Using rats expressing the SOD1^G93A^ mutation, microglial cells were shown to express senescence-associated markers, namely, p16, and the loss of nuclear laminin B1 in the spinal cord [201]. Senescent microglia from these animals grown in vitro exhibited typical characteristics of SASP which might play key roles in facilitating neuroinflammation. Overall, these studies suggest that any senescence-associated impairments in the innate immune cells alter glia-neuron communication, in turn promoting neuronal dysfunction and degeneration.

The direct evidence for the role of glial cell senescence in regulating the NLRP3 inflammasome, and its contribution to ALS, remains to be determined. It is important to note that normal physiological aging is associated with the appearance of systemic low-grade chronic inflammation, termed “inflammaging” [202]. Thus, during inflammaging, lifelong exposure to stressors is associated with chronic activation of the NLRP3 inflammasome [203], and it is plausible that this process is accelerated in ALS.

### 4.6. The Current Landscape of ALS Therapeutics

There are no effective treatments to slow or halt the progression of this devastating disease. Currently, ALS patients receive palliative treatment with riluzole, a glutamate antagonist, approved for use by the FDA in 1995, and edaravone, an ROS scavenger and an inhibitor of lipid peroxidation, was recently approved by the FDA in 2017 [204]. Nevertheless, these treatments provide only modest effects and only in a subset of patients; ALS patients are managed largely by providing supportive care [205].

Over the years, a large number of experimental therapeutics targeting inflammation have been tested in cellular and animal models of ALS through means of gene therapy, exon skipping; and pharmacological interventions such as mitochondrial protectants; antioxidants and anti-inflammatory compounds [206]. For instance, thalidomide, an inhibitor of TNF-α, reduced serum proinflammatory cytokine concentrations, lowered motor neuron death and improved motor function in an ALS mouse model; however, it did not meet clinical endpoints in a phase 2 clinical trial in a small cohort of ALS patients [207]. In fact, the results of more than 50 randomized controlled trials carried out in the past half-century for several disease-modifying treatments have been disappointing and failed to meet clinical endpoints [208].

The main barrier to drug development is the lack of understanding of the precise mechanisms involved in motor neuron degeneration. Dysregulation of a large number of cellular pathways has been implicated in ALS disease progression, and identifying the exact causative pathway remains a challenge. Most of the current knowledge on the mechanisms of ALS disease initiation and progression is derived from human genetics, post-mortem pathological tissues and animal models. These models have numerous limitations, as they do not recapitulate human ALS pathogenesis. An important factor is age, which is missing from the animal models of neurodegenerative diseases. The roughly 2-year lifespan of a rodent is unable to recapitulate the complex cellular and molecular processes of a human lifespan. Hence, there is a need for the development of better models that incorporate the element of age and improve our understanding of the neurodegenerative processes. Another obvious reason for the failed clinical trials is the lack of an objective biomarker to monitor whether the drug has actually engaged its target to have a pharmacodynamics effect. Hence, the present clinical trials conducted without an ALS biomarker are unable to distinguish between responders and non-responders, which may be attributed to ambiguous evaluation of the tested therapeutics. Therefore, there is a critical need for a more detailed and systematic study of the disease pathogenesis to advance our understanding of ALS and incorporate biomarkers in future clinical trials.

Human induced pluripotent stem cell (iPSC)-based disease modeling has presented itself as an efficient tool with which to model neuromuscular disease in vitro [209], and has become especially important for neurological investigations, as access to the affected cells is challenging [210]. Recently, an iPSC-based drug screening platform was used to identify novel ALS therapeutics [211]. Indeed, using iPSC-based technologies, three drugs (ropinirole, a dopamine D2 and D3 receptor agonist; retigabine, a neuronal Kv7 channel activator; and bosutinib, a tyrosine kinase inhibitor) were identified as potential ALS treatments and are under investigation in clinical trials [212,213,214]. Thus, iPSC technologies could provide efficient models for investigating the NLRP3 inflammasome in neuromuscular diseases as well as developing novel therapeutics targeting its assembly in these disorders.

## 5. Other Rare Neuromuscular Diseases

The NLRP3 inflammasome has also been shown to be involved in other rare neuromuscular diseases, such as Charcot–Marie-tooth (CMT) neuropathy and Myasthenia Gravis, which elicit symptoms of muscle weakness and muscle atrophy. The hereditary motor and sensory neuropathy CMT diseases are a heterogeneous group of genetic disorders characterized by damaged peripheral nerves and weak atrophied muscles, usually affecting the legs, feet, arms and hands [215]. Scientists have generated a variety of animal models expressing monogenic mutations that closely resemble the CMT phenotype [216]. Many CMT patients suffer from conduction velocity decay and hypertrophic demyelination. In addition, these patients also suffer from acute, subacute or chronic inflammation along with oxidative stress (i.e., ROS) that is involved with progression of the disease [217,218,219,220,221]. Interestingly, melatonin treatment in CMT neuropathy patients reduces oxidative damage and normalizes plasma proinflammatory cytokines such as IL-1β [217]. This reduction in inflammasome activation may correlate with reduction of the degenerative process.

Myasthenia Gravis (MG) is an autoimmune disorder that affects the postsynaptic region of the neuromuscular junction. More specifically, this disease is caused by specific autoantibodies, produced by B-lymphocytes, against the acetylcholine receptor (AChR), which is responsible for the transmission of signals between neurons and muscles [222,223]. Thus, mice models induced with an immune response to AChR represent excellent models with which to investigate the pathogenic mechanisms underlying the human disease [224]. Similarly to MG patients, experimental autoimmune MG (EAMG) mouse models show weakness in the skeletal muscles over time potentially, partly caused by the translocation of the neuronal NO synthase from the muscle membrane to the cytoplasm, due to blocked neuromuscular transmission [225]. Due to the suspected involvement of the NLRP3 inflammasome in autoimmune diseases [226], researchers were interested in investigating its role in MG. Overall, a caspase-1 inhibitor significantly ameliorated the symptoms of the disease in a EAMG rat model [227]. In addition, IL-1β expression levels were significantly increased in the serum of a EAMG mouse model [228]. However, the specific mechanism by which inflammasomes are activated has yet to be investigated. Since it has been shown that NO can induce IL-1β [229], it would be significant to determine whether the accumulation of NOS (responsible for the production of NO) in the cytoplasm of muscle cells would induce NLRP3 inflammasome activation. Thus, these rare neuromuscular diseases could indeed benefit from NLRP3 inflammasome-targeted treatments [217,227,228].

Other rare skeletal muscle disorders that are characterized by chronic muscle inflammation are termed inflammatory myopathies. Patients affected by these disorders suffer from inflammation and chronic or subacute progressive weakness of muscles and fatigue, which are often accompanied by muscle pain. These rare myopathies, often considered as autoimmune disorders, can be classified into five groups: polymyositis (PM), dermatomyositis (DM), necrotizing myopathy (NM), overlap myositis (OM) and inclusion body myositis [230]. Although glucocorticoid treatment is the basic therapy used to reduce symptoms for DM, PM, NM and OM, its beneficial effects vary from patient to patient, and it is associated with many long-term side effects such as osteoporosis, cataracts, hypertension, weight gain and increased risk of infections [230,231]. The development of therapeutics for inflammatory myopathies has been difficult since the immunopathogenic mechanisms underlying these disorders still remain poorly understood [232]. Previous studies supported the notion that inflammasomes might be involved in the pathogenesis of inflammatory myopathies. For instance, it has been shown that IL-1β and IL-18 promote the initiation and progression of PM and DM [233,234,235]. More recently, Yin et al. showed increased expression of the NLRP3 inflammasome in the muscles of DM and PM patients. In addition, increased expression of the NLRP3 inflammasome was associated with the activation of caspase-1 and enhanced secretion of IL-1β and IL-18 [232]. Another study has demonstrated that a metabolic switch to glycolysis in the lesioned DM and PM patients’ muscle tissues activates the NLRP3 inflammasome, causing pyroptosis of muscle cells [236]. Increased plasma IL-1β levels were also seen in these patients, suggesting that IL-1β may be a potential biomarker for these myopathies. Altogether, these studies support the role of the NLRP3 inflammasome in inflammatory myopathies and the potential for the development of NLRP3 inflammasome-targeted therapies.

## 6. Concluding Remarks and Perspectives

Although inflammation has emerged as a main culprit in the initiation and progression of neuromuscular diseases, the precise mechanisms involved are far from being fully elucidated. However, evermore studies support the notion that failure to control the acute inflammatory responses to nerve or muscle injury or dysfunction may be a primary trigger leading to a cascade of cellular events that culminate in muscle degeneration and nerve cell loss. The hallmarks of chronic inflammation are an increased number of circulating immune cells and their infiltration at the sites of injured peripheral nerves, neuromuscular junctions and muscles, which contribute to the progression of tissue damage. The presence of activated NLRP3 inflammasomes and their components has been demonstrated in innate immune cells, including neutrophils, macrophages, mast cells and microglia; adaptive immune cells, such as T-lymphocytes; and non-immune cells, such as myocytes and astrocytes. Neuroinflammation is also recognized as a contributing factor in the initiation and disease progression in ALS, a multifactorial neurodegenerative disease. Accumulation of misfolded and aggregated TDP-43 protein within neural cells (motor neurons and surrounding glial cells) has emerged as a common link between sporadic and familial ALS. The TDP-43 and other ALS-linked proteins might damage several cellular processes, resulting in dysregulated cellular homeostasis, which can, in turn, trigger inflammation. Increased production of ROS, nitric oxide and proinflammatory mediators is a cellular response to misfolded aggregated proteins, which can lead to oxidative stress, redox dysregulation and amplification of inflammatory responses [147]. Figure 4 depicts the possible crosstalk between the innate immune cells and activation of the NLRP3 inflammasome, leading to the motor neuron cell death and muscle degeneration. Despite a large number of mechanisms that have been identified in muscle degeneration and nerve cell loss, none have proven to be the primary cause of the disease. There is much need for a deeper understanding of the biology of the pathogeneses and the molecular mechanisms that are activated early in the diseases in order to identify “druggable” targets and disease-modifying treatments for these devastating diseases.

Human iPSC technologies are emerging as useful platforms for disease modeling to study pathogenic mechanisms and discover novel therapeutics for neuromuscular diseases [211,237]. Indeed, patient-derived iPSCs are being used to create a “patient-in-a-dish” disease model to derive relevant cell types for testing potential therapeutics, paving the way towards personalized medicine. This approach allows drug screening in a dish prior to administration to patients and “bench-to-bedside” translation of potential therapies. Additionally, iPSCs may also be used to stratify patients with various phenotypes and guide future clinical trials for bringing improved therapies to patients. Since multiple cell types are involved in disease pathogenesis, future research efforts need to be focused on deciphering “disease-specific signatures” at single-cell resolution, and not only in neuronal cells but also in non-neuronal cells. The application of modern technologies, including single-cell RNA sequencing and spatial transcriptomics, to neuromuscular diseases, will allow to ascertain cellular vulnerability and cell-specific mechanisms during various stages of disease progression.

The vital roles of the NLRP3 inflammasome in neuromuscular diseases such as DMD, LGMD and ALS, reveal that targeting this pathway is indeed a promising therapeutic strategy. Dysregulation of the NLRP3 inflammasome in muscle tissues by muscle damage, membrane instability, extracellular ATP and Ca^2+^ ions or signals from infiltrating immune cells, clearly impacts the progression of neuromuscular and neurodegenerative disorders. Thus, modulation of these pathways involved with activation and assembly of NLRP3 inflammasome could be truly beneficial. In addition, other neuromuscular disease patients (i.e., Charcot–Marie-tooth neuropathy and Myasthenia gravis) which elicit symptoms of muscle weakness and muscle atrophy also could benefit from NLRP3 inflammasome-targeted therapies [217,227,228]. Many approved therapies (i.e., canakinumab, anakinra and rilonacept) targeting the inflammasome in autoinflammatory diseases neutralize downstream IL-1β or compete with its receptor [238]. However, more directed therapies against upstream components of the inflammasome are being explored and tested, such as inhibitors, in neuromuscular and neurodegenerative models, as summarized in Table 1. These small molecules are also being optimized to cross the blood–brain barrier, which could be crucial for the treatment of neurodegenerative diseases. Thus, continued development of these small molecules may represent an important therapeutic approach to treating NLRP3-driven neuromuscular and neurodegenerative diseases. Specific targeting of the NLRP3 inflammasome components using antibody-based therapeutics is also on the horizon. In summary, based on the mounting body of evidence presented above, NLRP3 inflammasome inhibition has the potential to curtail aberrant inflammatory signaling and extinguish the fire of inflammation.

## Figures and Tables

**Figure 1 ijms-22-06068-f001:**
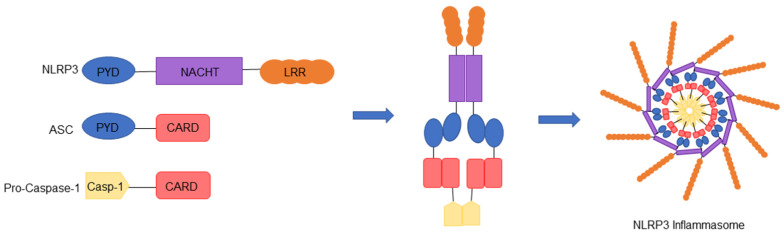
Assembly of the NLRP3 inflammasome. When activated, the adaptor protein (ASC) binds to NLRP3 via its pyrin (PYD) domain and creates a link to pro-caspase-1 via its CARD domain. The NLRP3 inflammasome is formed of NLRP3–ASC–pro-caspase-1 complexes interacting together to form a ring-like structure.

**Figure 2 ijms-22-06068-f002:**
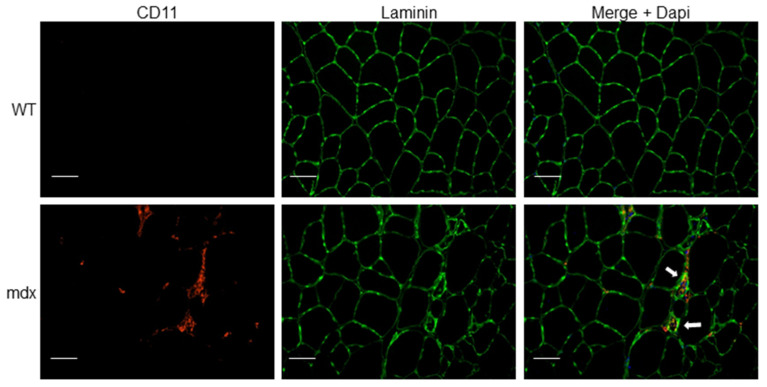
Immune cell infiltration in dystrophic muscle tissue. Tibialis anterior (TA) muscle cryosections from wild-type (WT) and mdx mice were immunostained with anti-CD11b (myeloid cell marker) and anti-laminin (muscle fiber membrane stain) antibodies. Nuclei were counterstained with 4′,6′-diamidino-2-phenylindole (DAPI, blue). Arrows indicate myeloid cell infiltrate (red) associated with focal invasion of muscle fibers (green). Scale bars, 50 μm.

**Figure 3 ijms-22-06068-f003:**
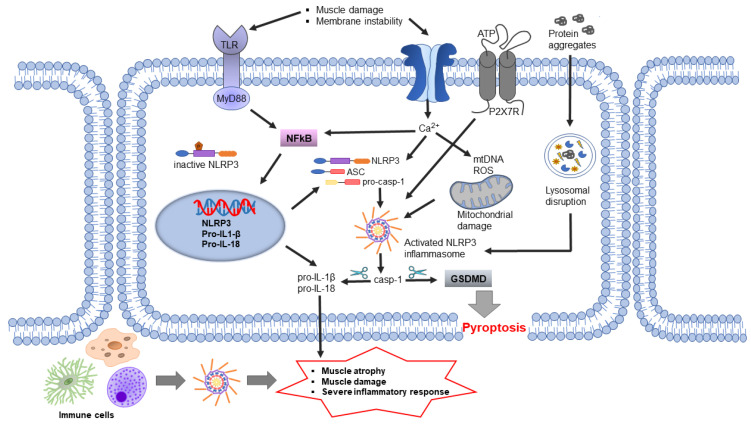
NLPR3 inflammasome involvement in muscular dystrophies. Disease-related effects via NLRP3 inflammasome activation in muscle tissues through membrane instability, extracellular ATP, Ca^2+^ ion influx or through signals from infiltrating immune cells.

**Figure 4 ijms-22-06068-f004:**
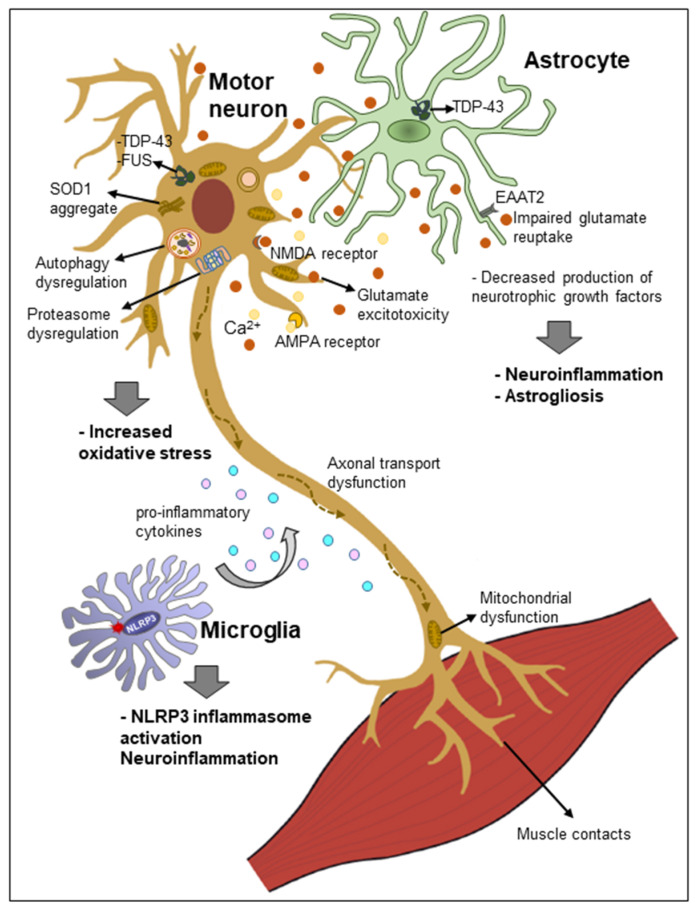
Proposed mechanisms of the pathogenesis of amyotrophic lateral sclerosis (ALS). The mechanisms underlying neurodegeneration in ALS are multifactorial and mediated through a complex interplay of molecular and genetic pathways. Specific pathways involved are increased generation of reactive oxygen species, glutamate excitotoxicity, mitochondrial dysfunction, axonal transport dysfunction and accumulation of cytoplasmic protein aggregates consisting of SOD1, TDP-43 and FUS. Activation of astrocytes and microglia results in secretion of proinflammatory cytokines, leading to neuroinflammation and motor neuron degeneration.

**Table 1 ijms-22-06068-t001:** Therapeutic targeting of the NLRP3 inflammasome in neuromuscular diseases.

Drug Type	Molecules/Drugs	Mechanism of Action	Disease	Model	Beneficial Effects	References
Small molecule	Prednisolone	NF-κB inhibitor	DMD	mdx DMD patients C57BL/6 mouse	- improves muscle strength and fiber integrity - prolongs ambulation and improves muscle strength - inhibits NLRP3 expression - reduces cleaved-caspase-1 and cleaved-IL-1β	[61,74,75,76,77,88]
	Edasalonexent * (formerly CAT-1004)	NF-κB inhibitor	DMD	mdx/GRMD dog **	- improves sarcolemmal integrity potentially via dysferlin and fibrosis - improves ex vivo function of the EDL and diaphragm - improves cardiac hypertrophy and cardiac fibrosis, diaphragm fibrosis and ventilatory - induces larger muscle volume	[239]
	VBP15 *	TNFα, NF-κB inhibitors	DMD	mdx	- improves muscle strength - promotes sarcolemmal repair of skeletal muscle cells	[79,240]
	MCC950	NLRP3 inhibitor	ALS	Primary mouse microglia TDP43 cell models	- attenuates IL-1β secretion via NLRP3 inflammasome	[135]
	Anakinra	recombinant human IL-1RA	ALS	SOD1^G93A^ mice ALS patients	- prolongs lifespan - improves motor performance (hanging wire test) - No significant effect	[180,189]
	GNX-4728 *	mitochondrial permeability transition pore (mPTP)	ALS	(hSOD1)^G37R^ tg mice	- protects mitochondrial degeneration - reduces inflammation, - extends lifespan	[241]
	Trehalose	Autophagy	ALS	SOD1^G93A^ mice	- protects mitochondria - reduces skeletal muscle denervation and motor neuron loss - decreases SOD1 aggregation - Increases autophagy - extends lifespan	[179]
	MitoTEMPO	angiotensin II	Muscle atrophy	C2C12 myotubes	- decreases mtROS and mitochondrial damage - inhibits NLRP3 inflammasome activation - increases expressions of p-PI3K, p-Akt, and p-mTOR - restores skeletal muscle atrophy	[47]
	Ibrutinib (PCI-32765)	NLRP3 inhibitor	Stroke	THP-1-differentiated macrophages C57BL/6J mice	- inhibits maturation of IL-1β - suppresses caspase-1 in macrophages and in the infarcted area of ischaemic brain. - Reduces ischaemic brain damage effects after stroke onset	[242]
Biologics	Adiponectin	miR-711	DMD	C2C12 myotubes/mdx	- decreases inflammation and oxidative stress involved in dystrophic muscle damage - decreases NLRP3 expression - reduces CK serum levels - improves physical performance	[13,93]
	Ghrelin	Suppression of JAK2/STAT3	DMD	Mdx	- reduces levels of NLRP3, ASC, pro-caspase-1, cleaved caspase-1, pro-IL-1β and IL-1β - improves behavioral performance and muscle fiber morphology	[83]
	cyclo (His-Pro)	NF-κB	ALS	(hSOD1)^G93A^ microglial cells	- reduces inflammation via NLRP3 inflammasome - decreases ROS	[190]

* Not yet tested for modulation of the NLRP3 inflammasome. ** Golden Retriever muscular dystrophy (GRMD).

## Data Availability

Not applicable.

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
