# Peer review of "Aberrant NLRP3 Inflammasome Activation Ignites the Fire of Inflammation in Neuromuscular Diseases"

_ijms, 2021, doi:10.3390/ijms22116068_

Round 1
Reviewer 1 Report
Excellent and timely review. Two minor suggestions/corrections: 1. Current survival with Duchenne muscular dystrophy is considerably longer than indicated with standard supportive therapy, and should be updated. 2. In the legend to figure 3 change neuromuscular dystrophies to muscular dystrophies. A somewhat larger concern regards the discussion of ALS, specifically what seems to be perhaps an excessive amount of attention to SOD1 FALS. To start with, although this was the first important gene associated with ALS, and opened up this very interesting field of oxidative stress and inflammation, it is important to point out that C9orf72 related ALS is far more prevalent both as familial and sporadic cases. It would be a nice improvement if this could be clarified, and this part of the paper reorganized to maintain clarity with respect to C9orf72, SOD1, familial and sporadic ALS.
Reviewer 2 Report
This is a very well-written review on the role of NLRP3 inflammasome in neuromuscular diseases. The NLRP3 inflammasome is a key player for sterile inflammation. Growing evidence suggests that innate immune system (including inflammasomes) is critically implicated in neurological diseases. This review clearly summarize the role of inflammasomes in neuromuscular diseases such as ALS.
Minor suggestions.
# 1. ALS immunopathology: A recent Cell paper showed the link between TDP-43 and cGAS/STING in ALS (yu et al. Cell 2020). This should be discussed.
#2. One missing topic is inflammatory myopathy (which should be directly linked with the Nlrp3-driven inflammation).
PKM2-dependent glycolysis promotes skeletal muscle cell pyroptosis by activating the NLRP3 inflammasome in dermatomyositis/polymyositis (Lieu et al. Rheumatology 2021)
The Familial autoinflammatory diseases (such as CAPS) will represent the bona-fide Nlrp3-drive muscular pathology (mainly fasciitis). Neuromuscular manifestations can be discussed. Ref. Neurology. 2010 Apr 20;74(16):1267-70.
#3. Table 1.
MCC950 is recognized as an inhibitor for NLRP3-mediated ASC oligomerization. No effect on NLRC4 or AIM2 activation (Coll et al. Nature Medicine 2015). Thus, it is better to describe as a Nlrp3 inhibitor, rather than ASC inhibitor.
Bruton's tyrosine kinase (BTK) inhibitor is also recognized as a Nlrp3 inhibitor, with neuro-protective effects post-stroke (Ito et al. Nat Communication 2015). This should be included.
